# Scale-Invariant Localization of Electric Vehicle Charging Port via Semi-Global Matching of Binocular Images

**Taoyong Li** [1,†], **Chunlei Xia** [2,†], **Ming Yu** [3,*], **Panpan Tang** [1], **Wei Wei** [2] and **Dongmei Zhang** [1]

1   Beijing Engineering Technology Research Center of Electric Vehicle Charging, Battery Swap, Beijing 100194, China; 15010311215@126.com (T.L.); tpp0739@126.com (P.T.); 15010099177@139.com (D.Z.)
2   Beijing Yupont Electric Power Technology Co., Ltd., Beijing 100029, China; study_xia@sina.com (C.X.); ravecn@163.com (W.W.)
3   School of Engineering, Beijing Forestry University, Beijing 100083, China
*   Correspondence: yuming@bjfu.edu.cn; Tel.: +86-10-6233-6221
†   These authors contributed equally to this work.

**Abstract:** Automatic charging for electric vehicles has broad development prospects for meeting the personalized service experience of users while overcoming the inherent safety hazards. An identification and positioning approach suitable for engineering applications is the key to promoting automatic charging. In this paper, a low-cost, high-precision method to identify and position charging ports based on SIFT and SGBM is proposed. The feature extraction approach based on SIFT is adopted to produce the difference of Gaussian (DOG) for scale space construction, and the feature matching algorithm with nearest-neighbor search, which is a kind of machine learning, is utilized to yield the map set of matching points. In addition, the disparity calculation is conducted with a semi-global matching algorithm to obtain high-precision positioning results for the charging port. In order to verify the feasibility of the method, a complete identification and positioning experiment of charging port was carried out based on OpenCV and MATLAB.

**Keywords:** identification and location of charging port; SIFT feature extraction; nearest neighbor search feature matching; semi-global matching method; disparity calculation



## 1. Introduction

The rapid development of new energy vehicles in the context of carbon peak and neutrality goals provides a new opportunity for the electric vehicle charging facility industry. With the increasing number of electric vehicles, the charging facilities for electric vehicles (EV) are gradually improving [1]. At present, processes of EV charging, such as charging port docking and charging time control of electric vehicles in the charging station, need to be completed manually; a large number of complex damage problems to the charging port caused by improper operation. At the same time, in view of the long-term exposure of the charging pile to the outdoors, it is inevitable that the insulation will be damaged due to aging, and there is a potential electric shock safety hazard of manual operation. With the advent of the era of intelligence, electric vehicle users will inevitably pay more attention to the intelligent and humanized experience of services [2–4]. Therefore, there is great value in studying the generalized automatic identification scheme of charging parts based on image recognition and to further design the automatic charging control system.

In recent years, a large number of studies related to automatic charging have been carried out. For example, Tesla has developed a serpentine high-degree-of-freedom charging machinery and equipment, but it is limited to the range of motion of the mechanical structure, which has high requirements related to the parking position of the vehicle, and is only suitable for Tesla models [5]. The E-Smart Connect system developed by Volkswagen in Germany uses sensors to trigger cameras to locate vehicles and interfaces, and the system controls the KUKA robot for charging [6]. Shi Ying designed a robot-based electric vehicle

charging system, which used binocular vision sensors to locate the coordinates of the charging port to drive the robotic arm, but did not conduct high-precision target positioning and experimental research under fuzzy light sources [7]. Zhang Hui from Hunan University applied machine vision to detect and locate charging ports, but did not obtain the pose in three-dimensional space, and only consider the case of round holes [8]. Sun Cheng studied the charging port identification and pose detection methods of electric vehicles under multiple disturbance factors, using monocular visual identification; although it is economical, it is not able to accurately obtain depth information [9].

High-precision identification and positioning algorithms are the top priority for automatic charging of electric vehicles, and an important prerequisite for ensuring the docking of charging devices. This paper proposes a high-precision charging port identification and positioning method suitable for different light intensities, backgrounds, and charging ports of any shape. First, the SIFT feature extraction algorithm is presented with the Gaussian difference pyramid being generated to construct the scale space. Then, the FLANN matching algorithm is utilized to obtain a high-precision mapping set of matching points. Second, in the process of binocular ranging, the SGBM (semi-global block matching) algorithm is used to calculate the parallax. This algorithm can calculate the parallax of the left and right camera images, so as to calculate the depth of the charging port more accurately. In order to verify the effectiveness of the proposed method, image identification and binocular ranging experiments were carried out, respectively, and high-precision matching and ranging results were obtained.

## 2. The Overall Process of Charging Port Identification and Positioning

First, to identify and locate the charging port through binocular vision, it is necessary to establish a binocular camera model. In order to solve the distortion problem in the perspective projection of the camera, camera calibration is conducted, through which the model parameters and distortion coefficients of the camera can be obtained for image correction. Second, based on the binocular vision system, the identification and localization algorithm of electric vehicle charging ports is studied, including filtering preprocessing of collected images, image segmentation, matching of pre-stored features of charging ports, and stereo matching of local graphic features. Finally, the three-dimensional spatial coordinates of the charging port are reconstructed; that is, the positioning of the charging port is completed [10]. The process of charging port image identification and positioning system is shown in Figure 1.

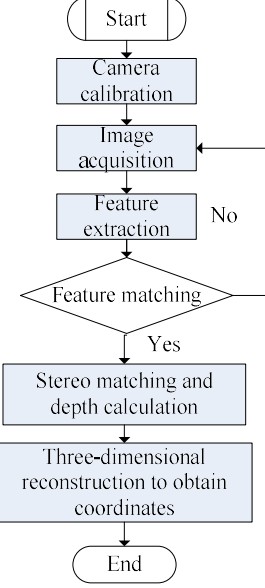

**Figure 1.** Process flow of charging port image identification and positioning system.

As the premise of identification and positioning of the charging port, the camera calibration process is based on the model of the existing camera. The parameters of the camera are calculated and transformed from the coordinates of the feature points. In the following, the three-dimensional reconstruction is carried out. Generally, "Zhang's calibration" method can be used, in which the value of the coordinate transformation matrix can be solved through more than four sets of points. However, in order to reduce errors and acquire stronger robustness, generally, many images should be taken and a large number of angle points selected for calibration [10].

## 3. Image Feature Extraction and Feature Matching Based on SIFT

The problem of charging port identification in the target area must be first solved to achieve automatic charging. The charging pile is generally placed outdoor with strong electromagnetic field, due to which the collected images are easily affected by noise. Therefore, the adopted charging port image identification method should possess good anti-interference performance in addition to strong robustness under different ambient light backgrounds. The target detection algorithm based on binocular vision is one of the most promising, as well as practical, methods in recent years, with its key lying in the feature point extraction and feature matching algorithm. SIFT (Scale-invariant feature transform), which is a scale-invariant feature transform algorithm, is a local feature detection method based on spatial scale extreme points. As the algorithm has invariance to rotation operation, scaling operation and brightness change in addition to strong robustness to noise and the characteristic of scalability, it can accurately extract the corner features in the image [11]. The extracted image features are then matched with the pre-stored charging port features to complete the identification function of the charging port. Figure 2 shows the flowchart of the image identification process of the charging port.

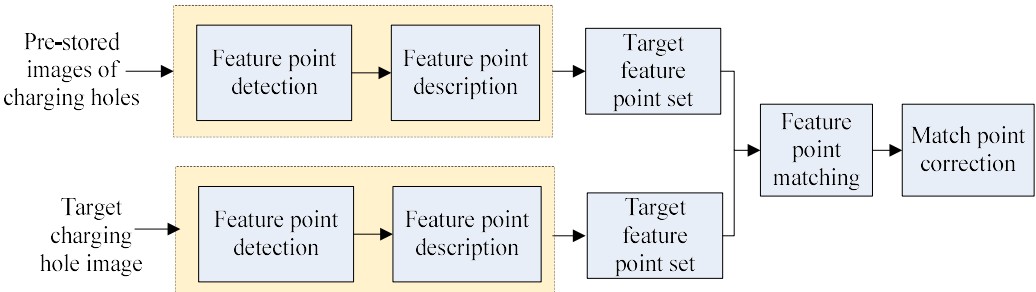

**Figure 2.** Schematic diagram of charging port feature extraction and matching.

The extraction of feature points by the SIFT algorithm can be realized by steps such as the construction of the scale space, the calculation of the spatial extreme points, the positioning of the stable key points, the information distribution of the direction of the stable key points, as well as the description of the key points which are shown as follows.

### 3.1. Scale Space Construction

Spatial scale coordinate transformation is performed on the detected image to obtain the scale space sequence. In the following, the main spatial contour of the scale space sequence is extracted, which is marked as a feature vector to complete the corner feature extraction of key points at different resolutions. The scale space constructed is invariant to scale changes, which is achieved by blurring and down-sampling the image through the Gaussian function [12–14]. In order to make the calculation relatively efficient, the Gaussian difference scale space is calculated by the Gaussian difference function, through which the Gaussian difference pyramid is generated, as shown in formula (1):

$$G(x, y, k\sigma) - G(x, y, \sigma) \approx (k-1)\sigma^2 \nabla^2 G \qquad (1)$$

In the formula, $x$ and $y$ are the scale coordinates. $\sigma$ is the image smoothness coefficient. $G(x, y, \sigma) = \frac{1}{2\pi\sigma^2} e^{-(x^2 * y^2)/2\pi\sigma^2}$ and $k - 1$ is a constant.

### 3.2. Finding the Extreme Point in Space

In order to find the extreme point of the Gaussian function, any pixel needs to be compared with its adjacent points in the image domain and scale space domain. In the two-dimensional space of the image, any pixel is compared with the adjacent 8 pixels, and in the scale space of the same group, the center pixel is compared with the 18 pixels of the adjacent layer image respectively. In this way, double local extreme points in the scale space as well as the two-dimensional space of the image can be obtained [15].

### 3.3. Precise Positioning of Stable Key Points

Noise and edges are prone to cause mutation of Gaussian values, so the local extreme points obtained in Section 3.2 need to be further confirmed and screened to remove unstable and falsely detected extreme points. In addition, a downsampled image is used when constructing the Gaussian difference scale space, and it is necessary to determine the exact position of the extreme points obtained in the image corresponding to the original image.

### 3.4. Stable Keypoint Orientation Information Assignment

The stable extreme point extraction based on different scale spaces ensures the scale-invariant characteristics. Whereas the distribution of the direction information of the key points, that is, the gradient of the extreme point, ensures the angular invariance and rotation invariance of the key points to the image. Define $L(x, y)$ as the original image space function, with the gradient magnitude of any key point being shown in formula (2), and the gradient direction in formula (3).

$$m(x,y) = \sqrt{(L(x+1,y) - L(x-1,y))^2 + (L(x,y+1) - L(x,y-1))^2}, \tag{2}$$

$$\theta(x,y) = tan^{-1}\left(\frac{L(x,y+1) - L(x,y-1)}{L(x+1,y) - L(x-1,y)}\right). \tag{3}$$

The direction of the key point is obtained through the gradient direction histogram. Firstly, the gradient direction of all pixels in the neighborhood of the key point is calculated with $10°$ being used as a unit direction interval for classification. Secondly, the number of key points that fall within each direction interval is accumulated and represented as a gradient direction histogram. Finally, the direction indicated by the maximum value of the longitudinal coordinate in the histogram is assigned to the key point as its main direction, and the direction with the number of key points equivalent to 80% of the peak value is utilized as the auxiliary direction of the key point [16–18]. The application of auxiliary directions can improve the robustness of the algorithm and help stabilize feature matching.

### 3.5. Description of Key Points

The key point description is the expression of the key point in mathematical language, which is a important step in realizing the matching of image feature points. It describes the key point and the surrounding pixels that contribute to it. The pixel area to be solved is first divided into blocks, and the gradient histogram of the corresponding block is then calculated to generate the direction vector. Therefore, the image information is expressed in an abstract form. As is shown in Figure 3, the gradient value of each block of pixels is Gaussian weighted to obtain eight orientations through which a 32-dimensional vector can be generated to be utilized as the mathematical expression of the key point. Experiments show that, for each key point, using a 128-dimensional vector descriptor to represent the key point can achieve the best effect.

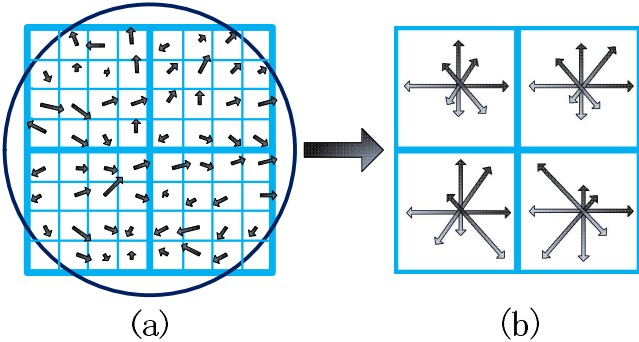

**Figure 3.** Description of key points: (**a**) image gradients around key points; (**b**) keypoint descriptors.

*3.6. Feature Matching*

After extracting the image features, it is necessary to perform feature matching with the pre-stored feature point set of the charging port to complete the identification of the charging port. Fast Library for Approximate Nearest Neighbors (FLANN) is a collection of nearest neighbor search algorithms for high-dimensional features in large data sets, optimizing the nearest neighbor search algorithms and high-dimensional features in large data sets. The FLANN matching algorithm records the feature points of the target image and the image to be matched, according to which a feature vector is constructed. By comparing and filtering the feature vectors, a mapping set of matching points is obtained. When using the FLANN matching algorithm, it is necessary to select an appropriate nearest neighbor search algorithm, such as random k-d tree algorithm, priority search k-means tree algorithm, hierarchical clustering tree, etc., as well as the number of recursive traversals. The more traversal times, the more accurate the results, but the longer the corresponding search time. Therefore, it is necessary to optimize and reasonably select the parameters [19].

**4. Binocular Ranging Algorithm**

According to the parallax theory, binocular vision positioning is based on the geometric relationship between the camera plane and the object to be recognized. The three-dimensional position information of the object is obtained through "feature matching" and "triangulation principle". In order to reduce the computational load and appropriately ease the matching difficulty, the images captured by the binocular camera need to be calibrated in the epipolar direction, making it an ideal binocular vision system. Through the stereo multi-dimensional matching technology, the correspondence between the points of the left and right images is determined to obtain the parallax, and then the depth and three-dimensional information of the image to be recognized is obtained according to the projection model. In general, stereo matching has always been a critical, yet difficult, question in stereo vision technology due to factors such as distortion, noise, specular reflection, and projection reduction.

The stereo matching method usually consists of four processes: the calculation of the matching cost, the aggregation of the cost, the acquisition of the disparity, and the refinement of the disparity. Among these steps, the calculation of the matching cost is the basis of the whole algorithm, which is the grayscale similarity detection under different parallaxes. Common detection indicators include the square or absolute value of the grayscale difference, corresponding to the different cost aggregation algorithms are adopted. After the matching costs are added, the disparity calculation of the local algorithm selects the minimum matching cost within a certain range as the matching point, while the global algorithm directly calculates the original matching cost to obtain the minimum value of the evaluation function. In addition, for some occasions with high-precision requirements, parallax refinement processing, such as image filtering, segmentation, and matching cost curve fitting, is also required.

SGM, known as semi-global matching algorithm, is represented modularly as the semi-global block matching (i.e., SGBM) in OpenCV. In the algorithm, the difference map

is constructed by selecting the difference of the pixel points, and the global cost function, which is related to the difference map, is set and minimized to solve the optimal difference of pixels [20]. The effect of SGBM stereo matching is better than that of local algorithms, but at the same time more complicated. The specific process of the algorithm is as follows:

(1) Preprocessing:

Step1: Use the horizontal Sobel operator for image processing.

Step2: Map the image pixels into a new image.

(2) Cost calculation:

Step3: The gradient cost of the preprocessed image is obtained by sampling method.

Step4: The SAD cost of the original image is obtained based on sampling method, and is superimposed with the gradient cost.

(3) Dynamic programming:

Step5: Establish a global Markov energy equation, and superimpose the full path information to calculate the pixel matching cost.

Step6: Add the multi-directional matching cost to obtain the total matching cost.

(4) Postprocessing:

Step7: Uniqueness detection.

Step8: Sub-pixel interpolation.

Step9: Consistency detection of the left and right images.

The SGBM algorithm attempts to establish a global Markov energy equation through the constraints of one-dimensional paths in multiple directions on the image. The final matching cost of each pixel is the superposition of all path information, and the disparity selection of each pixel is simply decided by WTA (Winner Takes All). The energy is accumulated in each direction according to the idea of dynamic programming, and then the matching costs in each direction are added to obtain the total matching cost, as shown in formula (4):

$$L_{\mathrm{r}}(p,d) = c(p,d) + \min \left\{ \begin{array}{c} L_{\mathrm{r}}(p-r,d) \\ L_{\mathrm{r}}(p-r,d\pm 1) + p_1 \\ \min\limits_{i=d_{\min},\dots,d_{\max}} L_{\mathrm{r}}(p-r,i) + p_2 \end{array} \right\} - \min\limits_{i=d_{\min},\dots,d_{\max}} L_{\mathrm{r}}(p-r,i) \quad (4)$$

In the formula, $L$ is the cost function accumulated by the current path; $P_1$ and $P_2$ are the smoothing penalties in the case of small and large differences in the disparity between the pixel and adjacent points with $P_1 < P_2$; the third term is adopted just to eliminate the effect caused by the difference in the lengths of each path in different directions. Furthermore, the total matching cost is obtained by adding up the matching costs in all $r$ directions. The penalty coefficient controls the smoothness of the disparity map; the larger the $P_2$, the smoother the disparity map.

## 5. Experimental Verification

In order to verify the feasibility and effectiveness of the proposed feature recognition and depth calculation of charging port image based on the SIFT algorithm and the SGBM algorithm, the image feature recognition experiment and binocular ranging experiment were carried out in turn. The binocular camera model DUAL-200M-030T160 is used in the experiment. Before acquiring the image of the charging port, the binocular camera needs to be calibrated and corrected first.

### 5.1. Camera Calibration and Stereo Correction Experiment

Considering factors such as calibration cost, accuracy, and robustness, the "Zhang's Calibration" method was selected to calibrate the binocular camera used in the project. First, a checkerboard is printed and stuck on the plane as a calibrator. Second, the orientation of the calibrator or camera is adjusted to obtain six pairs of calibration board pictures taken by left and right lenses respectively. Finally, the pictures are sent to the camera calibrator

calibration tool in MATLAB, from which the checkerboard corner points are extracted to estimate five internal parameters and six external parameters in the ideal distortion-free situation. Then, the actual distortion coefficient is further estimated using the least squares method for distortion correction (Figure 4) and stereo correction (Figure 5). After calibration and stereo correction of the camera, relevant experiments such as image feature extraction based on SIFT and feature matching can be carried out.

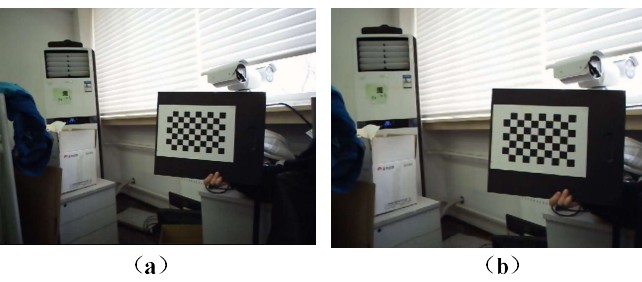

（**a**）　　　　　　　　　　　　（**b**）

**Figure 4.** Checkerboard before and after distortion correction: (**a**) Before distortion correction; (**b**) After distortion correction.

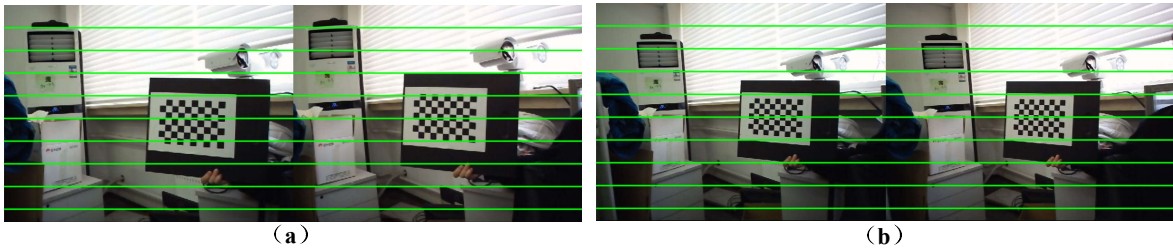

（**a**）　　　　　　　　　　　　（**b**）

**Figure 5.** Checkerboard before and after stereo correction processing: (**a**) Before stereo correction; (**b**) After stereo correction.

*5.2. Experiments on Image Retrieving and Key Point Extraction*

5.2.1. Image Preprocessing

The charging pile is generally set outdoors with a strong magnetic and electric scene, in which environment the image obtained by the camera will inevitably contain noise. Therefore, image filtering preprocessing is required to suppress the noise of the image while retaining the as many details of the image as possible to ensure the smooth extraction of image features. The commonly used filtering algorithms include mean filtering, median filtering, and Gaussian filtering. Performance comparison of the filtering methods is shown in Table 1, among which median filtering has the best effect on processing the noise of the charging port image and is adopted in this experiment. Figure 6 shows the feature points of filtering processed images.

**Table 1.** Performance comparison of filtering methods.

| Approach | Feature Points Extraction | Amount of Noise | The Influence of Noise on Feature Point Extraction |
|---|---|---|---|
| original image | Extract a large number of feature points | A lot | The interference is very large, and a large number of noise points are extracted |
| mean filtering | Extract many feature points | A lot, some being obvious | The interference is large, and more noise points are extracted |
| median filtering | Extract a large number of feature points | Very little, some not obvious | Less interference, a small number of noise points are extracted |
| Gaussian filtering | Extract a large number of feature points | A lot, some being obvious | The interference is large, and more noise points are extracted |

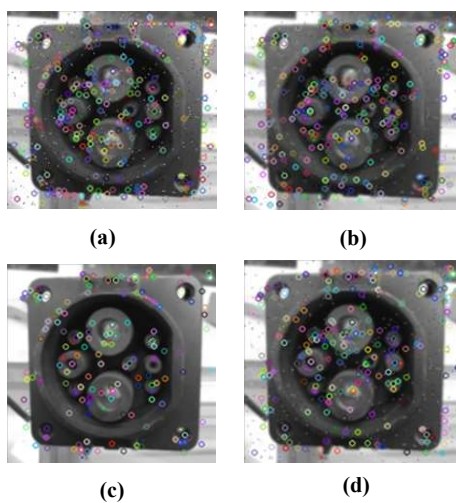

(a)      (b)

(c)      (d)

**Figure 6.** Feature points of filtering processed image: (**a**) before filtering; (**b**) by mean filtering; (**c**) by median filtering; (**d**) by Gaussian filtering.

### 5.2.2. Feature Extraction and Matching of Image

The image features of the charging port are extracted utilizing the SIFT algorithm in OpenCV. In order to enhance the robustness of feature matching, each key point is described by 16 seed points, generating 128 data points, which is a 128-dimensional SIFT feature vector that is not affected by scale changes and geometric deformations is finally formed. Further, normalizing the lengths removes the effect of lighting changes. The parameter settings are shown in Table 2.

**Table 2.** Parameters of SIFT algorithm.

| Parameter | Meaning | Value |
|---|---|---|
| nOctaveLayers | The number of levels in each group in the pyramid | 3 |
| contrastThreshold | Threshold for filtering out bad feature points | 0.04 |
| edgeThreshold | Threshold to filter out edge effects | 10 |
| double sigma | Gaussian filter coefficient of image in layer 0 of the pyramid | 1.6 |
| K(FLANN) | Top K points with the best match that the KNN algorithm returns | 1 |

After the SIFT feature vector is generated, the Euclidean distance method is used as the similarity criterion for key points. In order to exclude the key points with no matching relationship caused by background confusion or occlusion, a machine learning algorithm, K-nearest neighbor (KNN), is used to compare the nearest-neighbor distance and the next nearest neighbor distance. If the ratio is less than the set threshold, it is judged that the matching result is correct. By lowering the threshold, the number of matching points will decrease, but it will be more stable.

Figure 7 shows the features of the charging port extracted using the SIFT algorithm. It can be seen from the figure that the SIFT algorithm can extract the feature points of the charging port very well. After the extracted features are obtained, the FLANN matcher is utilized to perform feature matching between the feature points of charging port in the image extracted by SIFT and the pre-stored features, and the feature matching result is visualized as shown in Figure 8. It can be seen from the figure that the FLANN matcher can more accurately match feature points in the template extracted by SIFT with the target feature points.

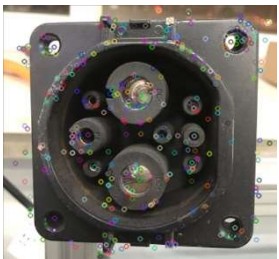

**Figure 7.** Feature points extracted by SIFT algorithm.

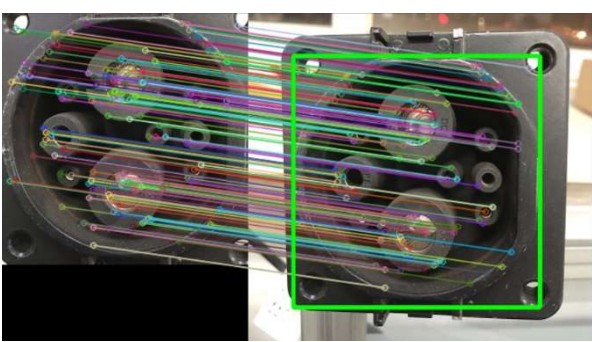

**Figure 8.** Feature matching results.

*5.3. Binocular Ranging Experiment*

After feature matching of the charging port is completed, the three-dimensional reconstruction of the charging port needs to be performed to identify the three-dimensional coordinates of the charging port. The disparity map can be obtained by using the BM algorithm and the SGBM algorithm.

5.3.1. Comparison of Parallax Map

Figure 9a is the disparity map obtained through the SGBM algorithm. The algorithm selects the difference of each pixel to form a difference map, related to which the global energy function is defined and minimized to obtain the optimal difference of each pixel. Figure 9b shows the disparity map calculated by the BM algorithm. The BM algorithm divides the frames of the two cameras into many small squares for model matching. By moving the small squares to match the small squares in the other image, and by finding the pixel positions of different small squares in the other image, combined with the relationship data of the two cameras (rotation matrix and translation matrix in the calibration parameters), the actual depth of the object is calculated to generate the corresponding depth map.

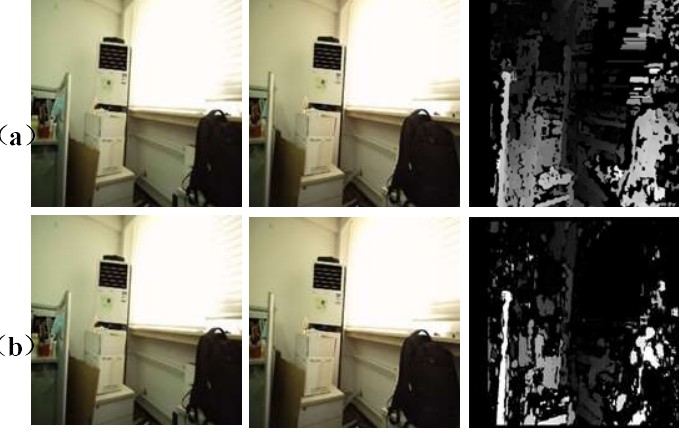

**Figure 9.** Comparison of disparity maps obtained by two algorithms: (**a**) SGBM algorithm; (**b**) BM algorithm.

Comparing the disparity maps obtained by the two algorithms, it can be seen that the disparity map processed by the SGBM algorithm is more refined than that of the BM algorithm, and with a better effect of the stereo matching. Therefore, the SGBM algorithm is used in this paper for stereo matching and disparity calculation.

### 5.3.2. Distance Measurement

Table 3 shows the results of multiple sets of ranging experiments on charging ports using the SGBM algorithm under different light intensities (1st to 6th rows: 20–500 lx; 7th row: 100,000 lx). The first six groups of data in the table are the test results of the binocular camera under randomly changing light within a small interval, and the last group is the test data under overexposure. The first three columns of the test results are the world coordinates (unit: mm) of the charging port relative to the left camera, and the three values are X (the horizontal direction of the captured image), Y (the vertical direction of the captured image), and Z (the distance between the camera and the charging port, i.e., depth). The fourth column is the actual distance of Z axis ($Z_A$) and the last two columns are the calculated overall distance (CD) and the actual overall distance (AD), respectively. It can be seen from the table that under the current camera calibration state and slowly changing light, when the actual Z distance is 200 mm (the distance is measured by the scale), the test distance RMS error between the calculated overall distance and the actual overall distance of the first six rows is about 1.51 mm (0.755%), which is acceptable in practical applications (<2% requested).

**Table 3.** Binocular camera range results (unit: mm) under (1st, . . . , 6th rows) random change in light in normal range and (7th row) an overexposure condition.

| X | Y | Z | $Z_A$ | CD | AD |
|---|---|---|---|---|---|
| −13.8329 | 4.0394 | 198.3303 | 200 | 198.8531 | 201 |
| −10.3983 | −32.383 | 199.3371 | 200 | 202.2179 | 201 |
| −20.1614 | 14.990 | 198.530 | 200 | 200.1133 | 201 |
| −13.5946 | 4.6332 | 200.5588 | 200 | 201.0724 | 201 |
| −14.6609 | 12.3290 | 201.5883 | 200 | 202.4964 | 201 |
| −19.4849 | 47.2711 | 200.7639 | 200 | 207.1722 | 205 |
| −57.71264 | 70.5011 | 283.3291 | 200 | 297.6181 | 220 |

It should be noted that the test error of the binocular camera is large under overexposure (the last set of data), which is also a disadvantage of binocular visual ranging. However, as the charging facilities are generally placed indoors (such as underground parking spaces) or are equipped with a rain cover or a roof, which is necessary for the automatic charging system of the research, the situation of overexposure can be excluded. Figure 10a is the visual ranging result and Figure 10b is the scale ranging image.

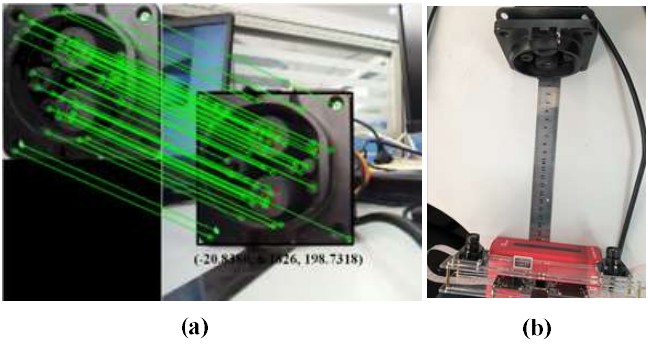

(**a**)                    (**b**)

**Figure 10.** Comparison of two measurement results: (**a**) Identification and distance measurement of charging port; (**b**) scale measurement results.

Tables 4 and 5 show the results of multiple sets of ranging experiments on charging ports when the Z axis is 250 mm and 300 mm, respectively (under different light intensities from 20 lx to 500 lx). $X_A$, $Y_A$, and $Z_A$ represent the actual distances of corresponding axes in each test. $|e|_I\%$ means the error between the measured value and actual value of axis I (I = X, Y, Z) and $|e|\%$ is the error between the calculated overall distance (CD) and the actual overall distance (AD). Furthermore, the mean error of each axis is illustrated by $|e|'_I\%$ (I = X, Y, Z) in the table below, from which it can be seen that although the errors in the X and Y axes are slightly bigger, the overall distance errors $|e|'$ are 0.71% and 0.60% in Tables 4 and 5, respectively, which are still in the acceptable range.

**Table 4.** Binocular camera range results when Z = 250 (unit: mm).

| X | $X_A$ | $|e|_X\%$ | Y | $Y_A$ | $|e|_Y\%$ | Z | $Z_A$ | $|e|_Z\%$ | CD | AD | $|e|\%$ |
|---|---|---|---|---|---|---|---|---|---|---|---|
| 47.57 | 50 | 4.86 | 21.44 | 20 | 7.20 | 251.59 | 250 | 0.64 | 256.94 | 256 | 0.37 |
| −8.89 | −10 | 11.10 | −78.19 | −80 | 2.26 | 249.81 | 250 | 0.08 | 261.91 | 263 | 0.41 |
| −42.99 | −45 | 4.47 | −68.37 | −70 | 2.33 | 249.06 | 250 | 0.38 | 261.83 | 263 | 0.44 |
| −80.62 | −80 | 0.78 | 68.15 | 70 | 2.64 | 249.55 | 250 | 0.18 | 270.96 | 272 | 0.38 |
| 44.53 | 45 | 1.04 | 22.82 | 23 | 0.78 | 250.56 | 250 | 0.22 | 255.51 | 255 | 0.20 |
| 34.80 | 35 | 0.57 | 15.79 | 15 | 5.27 | 245.23 | 250 | 1.91 | 248.19 | 253 | 1.90 |
| −32.28 | −30 | 7.60 | 78.66 | 80 | 1.68 | 249.55 | 250 | 0.18 | 263.64 | 264 | 0.14 |
| 4.37 | 5 | 12.60 | 3.84 | 5 | 23.20 | 243.23 | 250 | 2.71 | 243.30 | 250 | 2.68 |
| 23.26 | 25 | 6.96 | 63.40 | 60 | 5.67 | 247.57 | 250 | 0.97 | 256.62 | 258 | 0.53 |
| −4.27 | −5 | 14.60 | 63.64 | 65 | 2.09 | 249.06 | 250 | 0.38 | 257.10 | 258 | 0.35 |
| 47.80 | 50 | 4.40 | 67.67 | 70 | 3.33 | 249.30 | 250 | 0.28 | 262.71 | 264 | 0.49 |

Mean error: $|e|'_X = 6.27\%$ $|e|'_Y = 5.13\%$ $|e|'_Z = 0.71\%$ $|e|' = 0.71\%$.

**Table 5.** Binocular camera ranging results when Z = 300 (unit: mm).

| X | $X_A$ | $|e|_X\%$ | Y | $Y_A$ | $|e|_Y\%$ | Z | $Z_A$ | $|e|_Z\%$ | CD | AD | $|e|\%$ |
|---|---|---|---|---|---|---|---|---|---|---|---|
| −40.01 | −40 | 0.02 | 45.42 | 45 | 0.93 | 299.11 | 300 | 0.30 | 305.17 | 306 | 0.26 |
| −52.01 | −50 | 4.02 | −103.57 | −100 | 3.57 | 298.25 | 300 | 0.58 | 319.98 | 320 | 0.06 |
| −43.46 | −45 | 3.42 | −104.43 | −105 | 0.54 | 295.39 | 300 | 1.54 | 316.31 | 321 | 1.47 |
| 38.83 | 40 | 2.93 | 12.95 | 15 | 13.67 | 297.20 | 300 | 0.93 | 300.01 | 303 | 1.00 |
| 31.36 | 30 | 4.53 | −10.01 | −10 | 0.10 | 302.60 | 300 | 0.87 | 304.39 | 302 | 0.90 |
| −38.21 | −40 | 4.48 | 54.23 | 55 | 1.40 | 301.55 | 300 | 0.52 | 308.76 | 308 | 0.37 |
| −14.50 | −15 | 3.33 | 90.48 | 90 | 0.53 | 301.38 | 300 | 0.46 | 315.00 | 314 | 0.46 |
| 40.27 | 40 | 0.68 | −104.22 | −105 | 0.74 | 299.25 | 300 | 0.25 | 319.43 | 320 | 0.29 |
| −50.62 | −50 | 1.24 | −103.40 | −105 | 1.52 | 295.08 | 300 | 1.64 | 316.74 | 322 | 1.56 |
| −41.52 | −40 | 3.80 | −12.52 | −15 | 16.53 | 300.55 | 300 | 0.18 | 303.66 | 303 | 0.21 |
| −7.21 | −10 | 27.9 | 21.39 | 20 | 6.95 | 301.49 | 300 | 0.50 | 302.33 | 301 | 0.50 |

Mean error: $|e|'_X = 5.12\%$ $|e|'_Y = 4.23\%$ $|e|'_Z = 0.72\%$ $|e|' = 0.60\%$.

## 6. Conclusions

In this paper, a low-cost, high-precision identification and positioning method for charging ports suitable for engineering applications is proposed. This method adopts the binocular visual recognition technology, and deeply collaborative applications of the SIFT feature extraction algorithm, the nearest-neighbor search feature matching algorithm, and the SGBM disparity calculation method are conducted. Through operations such as camera calibration, scale space construction, spatial extreme point detection, stable key point position, direction information allocation, feature matching by machine learning and parallax calculation, etc., a charging port identification and positioning method suitable for different light intensities, backgrounds, and arbitrary shapes is obtained. In order to verify the feasibility of the method, a complete identification and positioning experiment of charging port was conducted. Through camera calibration and stereo correction experiments, the SIFT-based image recognition experiment, as well as the binocular ranging experiment, ideal identification of the charging port was obtained, providing a theoretical and technical foundation for subsequent research into charging docking driven by a robotic arm.

### 7. Patent

Automatic charging processing method and device for charging pile (No. 202111595712.9).

**Author Contributions:** T.L. proposed the algorithm; M.Y. and C.X. conceived and designed the experiments; P.T. and W.W. performed the experiments; M.Y. and D.Z. wrote the paper. All authors have read and agreed to the published version of the manuscript.

**Funding:** Supported by Open Fund of Beijing Engineering Technology Research Center of Electric Vehicle Charging/Battery Swap (China Electric Power Research Institute) (No. YDB51202101506).

**Institutional Review Board Statement:** Not applicable.

**Informed Consent Statement:** Not applicable.

**Data Availability Statement:** Not applicable.

**Conflicts of Interest:** The authors declare no conflict of interest.

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
