# Peer review of "Scale-Invariant Localization of Electric Vehicle Charging Port via Semi-Global Matching of Binocular Images"

_applsci, doi:10.3390/app12105247_

Round 1

Reviewer 1 Report

Authors propose a binocular visual recognition technology of electric vehicle charging hole based on SIFT and SGBM algorithms.
In my opinion it is a very practical proposal but it seems to me that there are some important explanation and experimentation data that needs to be added to the manuscript, such as:
3.3) At least a paragraph explaining the of the the global Markov energy equation to superimpose
the full path information to calculate the pixel matching cost and at least one example.
4.2.1 Statistichal comparison among the 3 filter approaches (not only visual) to conclude that the median filter has the best effect
4.3.1 SGBM algorithm meaning was explaind but not BM algorithm
4.3.2 The "different light intensities" values need to be explicitely informed.
"Z axis error" can be identified from Table 2 but not X and Y axis errors. 
And being Table 2 a very important Table for reader to understand the practicals result of the work it should have sufficiente information
a) intended exact position of robotic arm and X, Y and Z axis errors in each experiment 
b) inclusion of the light intensity in each experiment
c) finally and overall mean error for X, Y and Z axis in 50 to 100 tests indicating the range values min and max of the used light intensities.

Author Response

Dear editor and reviewers:

    The authors would like to thank you for your assessments and comments on the paper “Binocular visual recognition technology of electric vehicle charging hole based on SIFT algorithm” (Manuscript ID: applsci-1701224). We have carefully revised the paper and made amendments according to the requirements by the reviewers, and the words in red are the changes we have made in the paper. The detailed response is in the attachment.

Thank you very much again for all your help and looking to hear from you soon.

Best regards!

                                                               Sincerely yours

                                                                 Dr. Ming Yu

Reviewer 2 Report

Manuscript ID: applsci-1701224 
Title: Binocular visual recognition technology of electric vehicle charging hole based on SIFT algorithm  

Summary
This paper proposes a method of estimating the three-dimensional position of the charging socket of electric vehicles based on binocular camera images. The presented method is composed of a variety of the conventional vision techniques. The significance and the procedure of each step are carefully written. Experimental results using the proposed method are also presented.
    The theoretical and technical novelty of this work will be weak or not discussed with related works sufficiently. Nevertheless, the automatic detection of the charging socket is an important task of Computer Vision that has a social value and the authors’ work is worth sharing with readers of the Applied Sciences Journal.

Major concerns
[1] Technical terms and contribution of this work 
I would recommend organizing the technical terms again and polishing the manuscript based on what the authors actually did in this work. Concretely, I understand that the presented method is designed and examined to estimate the three-dimensional position of the charging port. However, the authors use ‘identification (identify)’ and ‘recognition’  in the title and the main text. It will be confusing to readers. 
    My understanding from the related papers is that: Identification is to estimate the presence or absence of the charging port in the given data; Recognition is to estimate the type of charging port within the known (pre-stored) charging ports; and Positioning is to estimate the location of feature points and 3D shape of the charging port. 
    The overview of the charging-port detection system (i.e., identification, recognition, and localization) can be still discussed as in Section 2. However, the authors should clarify their contributions in the paper title, Abstract, and the body text.

[2] Error measurement
(Section 4.4.2.) Please discuss the definition of error measurement in the revised manuscript. What is the estimated distance that corresponds to the actual distance? Is it Z coordinate? or the root sum square of (X, Y, Z) coordinates? I think the RSS should be calculated considering the practical use of the charging socket localization.

[3] Discussion 
In Line 310, the authors argue the distance error is ‘relatively accurate’. Please discuss more concretely. What does ‘relatively’ mean? Also readers will want to know whether the accuracy satisfies the requirement of the electric vehicle or a further improvement is required.

[4] Paper title
‘Technology’ in the paper title refers to a general method and knowledge.  I think ‘technique’ and ‘method’ are more suitable. Also ‘recognition’ can be misleading. Please reconsider the title. For example, ‘Scale-invariant localization of electric vehicle charging port via semi-global matching of binocular images’ would be a choice.

Minor concerns
Line 324, Table 2: How about modifying the caption as “Binocular camera ranging results (unit: mm) under (1st, …, 6th rows) random change of light in normal range and (7th row) an over-exposure condition.”  It will help readers understand the content of Table.

Line 172, Caption of Section 2.6.: What does ‘Eigenvalue’ mean? Please explain in the manuscript or modify the caption.

Line 244, 4.2 Experiment of Image Recognition Based on SIFT:
I think the recognition task was not tested in this work. The caption should be revised to ‘Experiments on Image Retrieving and Key-Point Extraction’ for example.

Line 292: finely blocks: I could not understand what this means. Is it “The BM algorithm divides the frames of the two cameras into blocks with fine scale for model matching,”?

Line 24, 0. Introduction: Sections should start from 1. Please check the template file.
Line 261: in the following table: -> in Table 1.  
Line 314: is equipped -> are equipped
Line 341: 6. Patents -> Patents

Best regards,

Author Response

(The authors gave the same response as above.)

Round 2

Reviewer 1 Report

Authors state that "After consulting a professional, it is learned that the method is qualified only when the recognition error is less than 2%".

Tables 4 and 5 need to be modified so they can show which tests present -individually- such accepted error rates.

Suggested modifications in Tables 4 and 5 are:

- Include 3 new columns, after "Actual X", "Actual Y" and "Actual Z"

with the titles |e|% (module of each error in percentage). Then calculate each module of X, Y and Z error for each test.

- X', Y' and Z' Mean errors should be stated as Module mean errors in %

Finally, comments about the results should be added or modified in order to explain cases that meet (or not) the required error rate less than 2%

Author Response

Dear editor and reviewers:

    The authors would like to thank you for your assessments and comments on the paper “Scale-invariant localization of electric vehicle charging port via semi-global matching of binocular images” (Manuscript ID: applsci-1701224). We have carefully revised the paper and made amendments according to the requirements by the reviewers, and the words in red are the changes we have made in the paper. The detailed response is in the attachment.

Thank you very much again for all your help and looking to hear from you soon.

Best regards!

                                                               Sincerely yours

                                                                 Dr. Ming Yu
